# Airway Ultrasound for Anesthesia and in Intensive Care Patients—A Narrative Review of the Literature

**DOI:** 10.3390/jcm11216327

**Published:** 2022-10-27

**Authors:** Alexandra Elena Lazar, Mircea Constantin Gherghinescu

**Affiliations:** 1Department of Anesthesiology and Intensive Care, George Emil Palade University of Medicine, Pharmacy, Science and Technology from Tirgu Mures, Gheorghe Marinescu Street no 38, 540142 Targu Mures, Mures County, Romania; 2Department of Surgery 1, George Emil Palade University of Medicine, Pharmacy, Science and Technology from Tirgu Mures, Gheorghe Marinescu Street no 38, 540142 Targu Mures, Mures County, Romania

**Keywords:** ultrasound, airway, difficult intubation, anesthesia, intensive care

## Abstract

Ultrasound is an everyday diagnostic tool. In anesthesia and intensive care, it has a role as an adjuvant for many procedures, including the evaluation of the airway. Ultrasound airway evaluation can help predict a difficult airway, visualize the proper positioning of an intubation cannula, or evaluate the airway post-intubation. Protocols need to be established for the better integration of ultrasound in the airway evaluation, however until a consensus is reached in this respect, the ultrasound is a reliable aid in anesthesia and intensive care.

## 1. Introduction

Ultrasound became a tool used by all specialists, regardless of their field of practice. There are very few medical fields where ultrasound is not used. Its non-invasive character and the possibility of visualizing deep structures and tissues allow specialists to provide better and safer care for their patients [1]. The ultrasound equipment is both suitable for emergent and scheduled procedures, although its exact role in emergencies is not well established [1].

Ultrasound devices are becoming more and more suitable for every procedure, and the producers have significantly improved the design and portability. These features, along with the possibility to connect the ultrasound to smart devices, allow the specialists to have their “pocket ultrasound” on hand every time, and most importantly, everywhere it might be needed [2].

However, possessing an advanced device is not the key to a correct image interpretation. A not-so-steep learning curve allows the specialists to choose the right window or transducer to obtain the best from the ultrasound machine [3].

Airway ultrasound is implemented more and more in the daily practice of clinicians, and it is proven to be useful even in some emergencies, such as finding the right position for performing an emergent cricothyrotomy [4].

The evaluation of the airway patency is based mainly on clinical signs and some paraclinical investigations. Their scope is to predict a difficult airway before any medical gesture upon the airway is performed, so the proper tools and maneuvers can be taken ahead by the specialists. Among the clinical signs used, is the Mallampati scale, the thyromental distance, the distance between the incisors, mouth opening, and neck mobility. Studies concluded that alone, these signs have a poor predictability of a difficult airway, but combined, their power of prediction increases [5,6,7,8]. However, these signs are subjective and prone to mistakes, hence a better assessment tool is sought for a difficult airway prediction.

The scope of this review is to point out the usefulness of the ultrasound in the airway evaluation in an anesthesia and intensive care setting.

## 2. Perioperative Ultrasound Airway Evaluation Utility

Difficult airway prediction still presents biases nowadays, and although the worse situation when the patient cannot be ventilated or intubated is rare, the impact can be disastrous [9]. The preoperative meeting with the anesthesiologist can uncover a few signs related to a possible difficult intubation, such as a reduced mouth opening, a reduced thyromental distance, the impossibility to bite the upper lip, or a high Mallampati grade [10]. However, these are prone to subjectivism and sometimes they do not describe the real situation which can be visible in a direct laryngoscopy, the Cormack–Lehane grading, or after the induction of the anesthesia, when the patient cannot be ventilated [8]. These are just a few situations where ultrasound can be of use, because it offers the possibility of an anatomy evaluation, which may uncover possible anatomical modifications that impede the proper ventilation or intubation.

## 3. Technical Aspects of an Airway Ultrasound

A quick ultrasound scan of the airway can be used to visualize the anatomy of the airway, before a laryngoscopy, to exclude possible difficult intubation. To obtain a good view of the airway, the recommended ultrasound probes are the linear low-frequency probes for structures closer to the skin, and the curved high-frequency probes are preferred for deeper situated structures, [11].

The proper patient position for an ultrasound airway evaluation is supine, with the neck extended.

The ultrasound airway identifiable structures of interest, for the anesthesiologist, are the mouth, tongue, the oro-, and hypopharynx, hyoid bone, larynx, vocal cords, cricothyroid membrane, cricoid cartilages, trachea, and esophagus. (Figure 1)

### 3.1. Ultrasound Approaches Which Are Practical for the Airway

Several ultrasonographic approaches are known and used to evaluate the airway: the transcutaneous ultrasound of the airway, the endobronchial evaluation, and the transoral or sublingual approach [12].

The transcutaneous approach comprises the trans laryngeal and transtracheal ultrasound windows used to evaluate the upper airway and can be performed using either a low- or a high-frequency transducer. The orientation of the probes can be one of the following: sagittal, parasagittal, oblique, or transverse [12,13,14] (Figure 1).

The endobronchial evaluation is a combination of the transcutaneous approach and bronchoscopy. It is more elaborate and requires complex equipment [14].

The transoral or sublingual approach is the least favored by both the anesthesiologists and the patients. It implies the insertion of the probe under the patient s tongue, which makes it uncomfortable for the patient. The great advantage of this approach is that provides excellent-quality ultrasound images, due to good probe-tissue contact [15].

### 3.2. Airway Measurements Performed by Ultrasound

Different distances and diameters or ratios between these structures are measured to identify the best one for predicting a difficult airway. Additionally, any anatomical anomalies or masses, if present, can also be identified with an ultrasound [16].

Although the anatomical structures that need to be visualized by the ultrasound window are well established, it is not clear which of these or which combination of diameters and distances are the best predictors for a difficult airway.

The literature presents different results with many ultrasound measurements proposed. Some of these are presented below.

Tongue thickness and anterior neck soft tissue, at the level of the hyoid bone, are measured from the skin on the short axis [17].Thyrohyoid membrane: from the hyoid bone to the thyroid membrane [17].Transverse tracheal air shadow diameter in the subglottic area of the front of the neck: from the mid-front part of the neck, at the level of true vocal cords [18].Soft tissue depth at the vocal cords and the suprasternal notch of the distance from the skin to the anterior part of the trachea, determined at three levels: vocal cords, thyroid isthmus, and suprasternal notch. The amount of soft tissue in each zone is considered the average of the amounts of soft tissue obtained in the central axis of the neck and ≈1.5 mm to the left and right of the central axis [19].The distance from the epiglottis to the midpoint of the distance between the vocal folds (E-VC) [20].The depth of the pre-epiglottic space and skin to the epiglottis [21,22].Submandibular ultrasound of the hyomental distance ratio: the hyomental distance measured in neutral and in the hyperextended position of the neck [23].Tongue volume is the result of the multiplication of the mid-sagittal cross-sectional area, by its width, in a transverse ultrasonogram [23].Visibility of the hyoid bone [21].Anterior commissure: the shortest distance from the skin to the anterior commissure [24].Tongue thickness and the ratio to thyromental distance: measured in the medial sagittal plane, after asking the patient to rest the tip of the tongue on their lower incisors, since the tongue is a dynamic muscle [25].

All of these diameters and ratios were measured with the patient in the supine position, except the hyomental ratio, which was measured in the resting position of the conscious patient.

## 4. Prediction of a Difficult Airway and Identification of Some Airway Pathologies, with the Aid of Ultrasound

Most of the literature presents the ultrasound of the airway as a possible method of detecting a difficult airway, but no exact measurements are proposed as being the ones to be measured before intubation. Just a few of the distances and diameters measured by ultrasound were found significant in a difficult airway prediction. These measurements were linked with the clinical signs of a difficult airway prediction, mostly with the Mallampati scale and the Cormack–Lehane scale, and the direct laryngoscopic view [23].

Some significant results were observed, regarding the tongue thickness prediction of a difficult airway—an increased thickness over 6.1mm was correlated with a difficult airway. Since the tongue is manipulated during an attempt at intubation, this result is of great importance when appreciating the airway [25]. The same authors found that the ratio of the tongue thickness to thyromental is a powerful predictor of a difficult airway.

On the same matter, the skin-to-epiglottic distance with a cut-off value of 2.54 cm (sensitivity 82%, specificity 91%) and the pre-epiglottic area cut-off value of 5.04 cm (sensitivity 85%, specificity 88%) were the best predictors of a difficult intubation [20]. Although the skin-to-epiglottic distance is the most studied index in an ultrasound predicted difficult airway, the cut-off values varied among researchers and further studies are necessary. One study points out that the main cause of this disparity is ethnicity and gender, the differences are up to 1 cm in measurements [26]. Other authors found differences from 1.6cm in the Indian population and 2.5 cm in Italians, when they measured the skin-to-epiglottis distance. These variances are assigned to the anthropometric difference among the different races, mainly due to the fat distribution on the torso—in Asians, the fat is primarily distributed on the trunk [20,26,27].

Ezri T. et al. obtained significant results regarding a difficult airway prediction with a 2.7 mm cut-off value of soft tissue measured at the vocal cords level, and a 4.3 mm soft tissue at the suprasternal notch [19].

These diameters and measurements, can be carried out at the bedside and can aid in the diagnosis of a possible difficult intubation.

### 4.1. Determination of the Trachea Size or the Tracheal Potential Abnormalities

Starting from the known normal values of the tracheal wall, 1.5 ± 0.2 mm in males and 1.2 ± 0.2 mm in females, the possible tracheal thickening can be identified by airway ultrasound. These situations are present in inflammations, infections, tumor infiltration, congenital diseases such as Wegener’s disease, smoke inhalations, or airway burns [28,29].

Appreciation of the inner and outer tracheal diameters helps to determine the appropriate cannula size. The ultrasound, by its non-invasiveness and portability, makes an excellent tool in this respect, as opposed to other imagistic methods, such as the X-ray, computer tomography, or magnetic resonance [30].

The subglottic view of the trachea can offer a proper measurement for choosing the correct cannula, simple or double lumen to avoid over - instrumentation of the airway. The trachea appears on the ultrasound as a curvilinear structure, hyperechoic, with artifacts such as comet tail and shadowing [31]. The diameter measured to determine the proper cannula size is the subglottic airway one and for the double lumen tube is the measurement of tracheal width at the suprasternal notch level [32,33] (Figure 1).

A method to avoid the artifact produced by the inflation of the cuff, which is air and is not visible on the ultrasound, is to use a saline test and inflate the cuff with saline. This way it can be visualized on the ultrasound: tracheal rapid ultrasound saline test (T.R.U.S.T.) [34].

### 4.2. Cuff Pressure and the Trachea Wall Pressure Determination

The inflation of the cuff of the tracheal cannula is important for sealing the airway for both protection and proper ventilation. Intensive care patients may benefit from clear results in this regard, due to a prolonged intubation often encountered in the intensive care units. Studies have shown that after the cuff inflation, the diameter of the trachea modifies, this modification is visible at the suprasternal notch ultrasound window [35]. In Ye et al. the study of both the inner and outer tracheal diameters, were strongly correlated with the cuff inflation pressure. An example from the same authors shows, in an experimental animal study, that a 10 mL inflation volume produces 7.55 % (5.39, 9.12) of the outer tracheal diameter and 11.20% (8.16, 14.90) of the inner tracheal diameter [30].

These findings reiterate the importance of ultrasound in this respect, especially because this type of investigation can be carried out in real-time, at the patient’s bedside, or right before the anesthesia is commenced.

While cuff inflation modifies the tracheal diameter and offers to seal, the tracheal wall pressure is applied by the cuff on the trachea. This can lead to ischemia or even pathological communication between the trachea and esophagus, also known as fistulae [36]. The cuff pressure is different from the tracheal pressure exerted by the cuff. To date, there is no exact method of determining this pressure. The proposed methods for tracheal wall pressure determination are the pressure difference, the wall pressure membrane technique: invasive and feasible only in vitro, and the microchip sensor probe technique which gives false high-pressure values [37,38]. The pressure difference is an estimation of the tracheal wall pressure, and it implies complex calculations between the inner and outer tracheal diameters and their modifications with the cuff inflation. There are favorable studies in this respect, with positive correlations between the above-mentioned measurements, but further analysis is needed to reach a clear conclusion [30].

### 4.3. Epiglottitis Diagnosis

An inflamed epiglottis can lead to a difficult ventilation and intubation. The airway ultrasound can detect such a modification. The best ultrasound window is the mid-sagittal transverse approach of the neck with a low-frequency probe. The diameter, which is significantly modified in epiglottitis, is the anterior-posterior diameter of the epiglottis [39].

Patients in whom neck radiotherapy was performed, will benefit from this evaluation because, in these patients, epiglottitis is frequently encountered as a radiotherapy side-effect [40].

### 4.4. Determination of Thyroid Cancer or the Goiter Influence upon the Airway

Intubation for thyroid surgery is known to pertain to a possible difficult intubation category because of the tumor or the goiter that might be upon the airway. An ultrasound scan of the upper airway of such patients might expose an abnormal airway and avoid the surprise of a very difficult airway which can lead to a situation of “cannot intubate, cannot ventilate”. The preferred windows for obtaining the best images in this situation are suprasternal and supraclavicular, with a low-frequency probe [41,42].

### 4.5. Vocal Cord Functionality

Vocal cords are visualized when the transducer is placed at the thyroid cartilage level. It can be tilted cranially or caudally for a better image. This investigation can evaluate the vocal cord mobility after thyroid surgery, after a prolonged intubation in intensive care patients, or before general anesthesia [43,44,45]. The advantages of this evaluation are that it is non-invasive, it does not cause discomfort for the patient, and it has high reproducibility, and is easy to learn and perform [46] (Figure 2).

## 5. Discussion

Ultrasound is a tool that is needed in the anesthesiologist and intensivist armamentarium. Since the technology is advancing and the devices become more accessible, in terms of prices and their dimensions, performing an ultrasound in the operating theatre or by the patient’s bed in the intensive care units is an everyday practice.

Airway ultrasound may not be employed as often as other ultrasound procedures, but it is proven that it has its advantages, especially in perioperative settings, when evaluating the airway before intubation, can be useful for the anesthesiologist to be prepared, in case of a difficult airway.

An ultrasound of the airway can have a purpose, even in the already intubated patient, when the proper positioning of the tube is verified or when one is interested in the tracheal wall pressure from the cannula’s inflatable cuff. This last aspect is specific to intensive care patients with prolonged intubation periods. This is still a grey area because there is no exact measurement approach for this pressure, to date. Knowing the tracheal wall pressure produced by the inflated cuff is important because it can lead to tracheal wall ischemia and even further to tracheal-esophageal fistula. Although further studies are needed in this direction, the ultrasound can still make a difference, when used to evaluate the potential damage from the inflated cuff.

Unfortunately, this type of evaluation is not so useful in emergencies, because it may be time-consuming, and the image interpretation is prone to subjectivism from the performer. The learning curve is not as steep as it will be needed to utilize it in an emergent situation. Studies on this aspect showed that the learning curve can be shorter for some structures—such as identifying the trachea, the cricothyroid membrane, and confirmation of the tracheal tube position—but longer for the calculation of different diameters or identifying possible abnormalities and interpreting the images. This is especially because the ultrasound examination is user-dependent and it has a low reproducibility of the images, as opposed to other imagistic evaluations [47,48].

This review of the literature reiterates the importance of ultrasound for airway management, from predicting a possible difficult intubation to maintaining a correct intubation, without any damage to the tracheal wall. Although there are many diameters and measurements which can be performed with the aid of ultrasound on the upper airway, just a few proved to have a high specificity in the difficult airway prediction. Even so, when the anesthesiologists encounter a patient with a possible difficult airway, perioperatively, they can visualize the airway before putting the patient under anesthesia and plan their procedures to increase the patient’s safety.

In conclusion, the ultrasound evaluation of the airway is of great importance and with the advent of technology, it can improve patients’ safety, regardless of the patient’s need for intubation or if it is already intubated. The ultrasound aids with the supervision of the effects of the intubation to prevent any side effects, such as fistulae or damage to the tracheal wall. Protocols are needed in this direction so the ultrasound evaluation of the airway can be more accessible to anesthesiologists and intensivists. Once these protocols and recommendations reach a common point, this medical evaluation will be more accurate, and error-free, and the patient’s safety will be increased. This review, with other research in the field, can constitute the backbone for protocol creation and possible implementation of the airway ultrasound point of care.

## Data Availability

Not applicable.

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
