# Peer review of "Airway Ultrasound for Anesthesia and in Intensive Care Patients—A Narrative Review of the Literature"

_jcm, 2022, doi:10.3390/jcm11216327_

Round 1

Reviewer 1 Report

Manuscript ID: jcm-1967017
Type of manuscript: Review
Title: Airway ultrasound for anesthesia and in intensive care patients – a 
narrative review of the literature

In this manuscript, the authors produced a narrative review of the literature about the role of “…Ultrasound use in Anesthesia and Intensive Care. The authors concluded that the ultrasound evaluation of the airway is of great importance and with the advent of technology, it can improve patients’ safety regardless of the patient’s need for intubation or if it is already intubated. The ultrasound aids with the supervision of the effects of the intubation to prevent any side effects, such as fistulae or damage to the tracheal wall…”

The produced manuscript is according to the standards of reporting the subject may be relevant and important, and the information provided is clear, and objective.

Being a narrative of the literature, it is not necessary to propose protocols but they conclude that “Protocols are needed in this direction so the ultrasound evaluation of the airway can be more accessible to anesthesiologists and intensivists.” I think that the authors could have gone beyond and proposed some protocols for fundamental aspects of airway management use of ultrasound like the prediction of a difficult airway and may propose a systematic approach to airway evaluation.

The manuscript is well written, and I have no further comments on the topics and structure that is well designed.

Author Response

Thank you for reading our work and revising it. Indeed, as the reviewer pointed out, we touched the line when suggesting that some protocols should be in place for this procedure. This work, along with others on the matter, may be the background for the proposed protocols. 

We added this sentence in this respect

"This review, with other research in the field, can constitute the backbone for protocol creation and possible implementation of airway ultrasound point of care."

Best regards, 

Alexandra Lazar

MD, PhD

.....

Reviewer 2 Report

Thank you very much for the opportunity to review the manuscript, the manuscript deals with an interesting and innovative topic, but I think that the presentation is not fluent enough and the manuscript is presented like a chapter in a book and not in a sufficiently interesting manner. There is a need for significant editing, reduction of duplication and addition of practical tools.

Some Comments:

1. Lines 20-22 - please add a reference.

2. Figure 1 - unnecessary, you should add along the manuscript which probe is recommended to be used for each exam. You only wrote it in some of the exams.

3. Throughout the manuscript, the authors repeat several times that the tests for evaluating a difficult airway are not specific and there is no one exact test. No need to state this multiple times.

4. Lines 148-151 - The authors note differences in ultrasound between different populations, do these differences also exist in exams that are not sonographic?

5. Are there any prospective studies that have tested the ability of ultrasound in predicting a difficult airway and the clinical correlation?

6. There are no practical recommendations that the authors offer mainly for emergency situations where there is no time for multiple tests.

Author Response

Dear Reviewer, 

Thank you for taking the time and revising our manuscript. Below are the answers to the observations which were made:

1. Lines 20-22 - please add a reference - the reference was added

2. Figure 1 - unnecessary, you should add along the manuscript which probe is recommended to be used for each exam. You only wrote it in some of the exams.- we excluded the figure 

3. Throughout the manuscript, the authors repeat several times that the tests for evaluating a difficult airway are not specific and there is no one exact test. No need to state this multiple times.- we excluded from the text several mentions regarding the inaccuracy of the method

4. Lines 148-151 - The authors note differences in ultrasound between different populations, do these differences also exist in exams that are not sonographic?- yes, differences between different human races might be in regards to their height, mouth opening, incisor length, and protrusion but these are not always predicting a difficult airway.

5. Are there any prospective studies that have tested the ability of ultrasound in predicting a difficult airway and the clinical correlation?- yes, we provide here two recent prospective studies on this matter. 

  • Kasinath MPR, Rastogi A, Priya V, Singh TK, Mishra P, Pant KC. Comparison of Airway Ultrasound Indices and Clinical Assessment for the Prediction of Difficult Laryngoscopy in Elective Surgical Patients: A Prospective Observational Study. Anesth Essays Res. 2021 Jan-Mar;15(1):51-56
  • Miguel Angel Fernandez-Vaquero, Pedro Charco-Mora, Miguel Angel Garcia-Aroca, Robert Greif,
    Preoperative airway ultrasound assessment in the sniffing position: a prospective observational study,
    Brazilian Journal of Anesthesiology, 2022, doi.org/10.1016/j.bjane.2022.07.003.

6. There are no practical recommendations that the authors offer mainly for emergency situations where there is no time for multiple tests. - this type of evaluation is not so useful in emergency situations, because it may be time-consuming, and the image interpretation is prone to subjectivism from the performer.

With the hope that our answers were satisfactory we sent our Best Regards

Alexandra Lazar

MD, PhD

Round 2

Reviewer 2 Report

Dear Editor,
the authors have revised the manuscript “Airway ultrasound for anesthesia and in intensive care patients – a narrative review of the literature“.

The authors have addressed points in detail and have commented on them. Due to the changes, the manuscript has improved in quality. I still think there is a need for style improvement in order to improve the writing fluency.

Author Response

Dear Reviewer, 

many thanks for the improvement ideas and your appreciation of our manuscript.

We modified some sections and rearranged them with the scope of style improvement. The modifications can be seen in the revised version we uploaded.

Regards

Alexandra Lazar

MD, PhD